# Effect of *Schisandra chinensis* Extract Supplementation on Quadriceps Muscle Strength and Fatigue in Adult Women: A Randomized, Double-Blind, Placebo-Controlled Trial

**DOI:** 10.3390/ijerph17072475

**Published:** 2020-04-04

**Authors:** Jinkee Park, Seoungho Han, Hyuntae Park

**Affiliations:** 1Department of Sport Rehabilitation, Dong Ju College, Busan 49318, Korea; park7166@gmail.com; 2Department of Family Medicine, College of Medicine, Dong-A University, Busan 49201, Korea; drhans@dau.ac.kr; 3Department of Health Care and Science, Dong-A University, Busan 49315, Korea; 4Laboratory of Smart Healthcare, Dong-A University, Busan 49315, Korea

**Keywords:** *Schisandra chinensis*, muscle strength, resting lactate

## Abstract

The fruit of *Schisandra chinensis* (SC) is a well-known traditional herb used for pharmacological purposes in Asian countries (e.g., Korea, China, and Japan). In animal studies, SC extract supplementation had beneficial effects on muscle strength and lactate level. However, the effect of SC extract supplementation on skeletal muscle strength and lactate at rest in humans remains unclear. The purpose of this study was to evaluate the effect of SC extract supplementation on quadriceps muscle strength (QMS) and lactate at rest in adult women. Forty five healthy post-menopausal middle-aged women (61.9 ± 8.4 years) were randomly divided into the SC (*n* = 24) or the placebo group (*n* = 21). The SC group consumed 1000 mg of SC extract per day, whereas the placebo group consumed 1000 mg of starch per day for 12 weeks. The difference in muscle mass, physical function, and biomarkers and the relative changes between baseline and 12 weeks were evaluated. We used two-factor repeated measures analysis of variance (ANOVA) to determine interaction (group × time) effects for variables. Statistical significance was accepted at *p* < 0.05. In ANOVA results, QMS (*p* = 0.001) and lactate level (*p* = 0.038) showed significant interactions. With paired t-tests, QMS was significantly increased (*p* < 0.001) and lactate level at rest was significantly decreased (*p* < 0.05) after 12 weeks in the SC group. However, no interactions were found between the other variables. Supplementation of SC extract may help to improve QMS as well as decrease lactate level at rest in adult women. We believe that SC extract is a health supplement that can support healthy life in this population.

## 1. Introduction

*Schisandra chinensis* (SC) extract has an extensive history as a medicinal herb in traditional Asian medicine for the treatment of diabetes mellitus, chronic cough, thirst, night sweating, hypertension, and obesity [1,2]. Several studies have reported that SC extract has effectively inhibited the damage of large arterial [3,4]. It has also been implicated in increasing endurance, accuracy of movement, and physical working capacity, as well as the control of arterial blood pressure [1] and the improvement of metabolic markers [2,5].

Loss of muscle mass and strength are predictors of all-cause mortality, coronary heart disease (CHD), stroke, hospitalization, and disability in the middle-aged and elderly population [6,7]. Muscle strength and quantity can be used as markers of muscle quality; however, muscle strength is more important when estimating mortality risk [8,9]. For example, decreased muscle strength was strongly associated with aging, protein-energy wasting, and physical inactivity in the middle-aged and the elderly [10]. Furthermore, decreased leg muscle strength, specifically, was associated with an increased risk of mobility loss, physical disability within instrumental activities of daily living (IADLs), and functional limitation [11,12].

Recent animal studies have reported beneficial effects of SC extract, for example, its supplementation appears to decrease protein degradation and increase protein synthesis, as well as display antioxidant and anti-inflammatory effects on skeletal muscle fibers [13,14]. Additionally, SC extract was also found to improve mass and strength in skeletal muscle, as well as endurance in mice [13,14,15]. However, in humans, SC extract may not be as effective as in animals. Our first hypothesis is that the supplementation of SC extract in humans can increase muscle strength. 

Blood lactate (BL) levels have been widely used as an indicator of muscle fatigue during the physical stress of the human body, such as acute exercise [16,17,18]. However, reliable cross-sectional and follow-up studies in adults have shown that high levels of resting BL are associated with a high risk of type 2 diabetes, CHD, and stroke [19,20]. 

Meanwhile, a recent study by Cao et al. reported that long-term SC supplementation could prevent increased lactate after exercise [15]. However, it is still unclear the effect SC extract has on lactate level at rest and on skeletal muscle strength in humans. Our second hypothesis is that SC extract supplementation in humans reduces lactate at rest.

Thus, we aimed to investigate the effect of SC extract on quadriceps muscle strength (QMS) and lactate level at rest, in healthy post-menopausal middle-aged women.

## 2. Materials and Methods

### 2.1. Participant Recruitment and Characteristics

This randomized, double-blind, placebo-controlled trial was performed in the Regional Clinical Trial Center, Dong-A University Hospital, Korea. The 65 subjects were recruited by poster advertisements in the hospital or community welfare center. The inclusion criteria for this study were as follows: healthy post-menopausal middle-aged women, with a body mass index (BMI) of 18.5–30.0 kg/m^2^, who provided written consent to participate in the study. Subjects were excluded if they suffered with any muscle strength-influencing disease, heart disease, or a chronic disease, such as diabetes or hypertension.

Additionally, subjects were deemed unsuitable and also excluded if they possessed an aspartate aminotransferase and alanine aminotransferase > 60 IU/L, were hepatitis C positive, or participated in regular physical activity programs or other clinical studies within the last 3 months. As a result of this, 13 subjects were excluded in this study. The 52 participants (55–78 yrs) were randomly divided into an SC group (*n* = 26) and a placebo group (*n* = 26), at a ratio of 1:1 by block randomization. 

During the 12-week period, 7 subjects dropped out due to the following: adherence rates of 2 subjects in the SC group were unclear, and 5 subjects in the placebo group withdrew consent (*n* = 3) or experienced symptoms of abdominal pain (*n* = 2). The final study consisted of 45 adult women (SC group = 24, placebo group = 21), confirmed as healthy via medical evaluation through a clinical interview with the family doctor and without chronic disease, cancer, heart disease, and cerebrovascular disease.

The study was approved by the Institutional Review Board of Dong-A University Hospital (DAUHIRB-14-147) and was registered in the UMIN Clinical Trial Registry (UMIN 000024169; https://center.umin.ac.jp/cgi-open-bin/ctr/ctr.cgi). Body weight and height were measured to the nearest 0.1 kg and 0.1 cm, respectively. BMI was calculated as weight (kg) divided by height squared (m^2^). Appendicular skeletal muscle mass (ASM) for the entire body was measured by dual-energy X-ray absorptiometry (GE Lunar DPX, GE Healthcare Technologies Lunar, USA) using an analyzer. The ASM/m^2^ was the ASM divided by height squared [21].

### 2.2. Physical Function Test

The physical function test used the maximum value, after taking measurements of five different physical parameters twice, between 8:00–10:00 a.m. These measurement parameters and methods are as follows: grip strength was measured using an isometric dynamometer (TKK-5401, Tokyo, Japan) with the dominant arm. The measurement of one leg standing duration was terminated when the lifted lower limb touched the floor, or the position of the supporting lower limb shifted. The walking test measured the time taken to walk a 4 m straight distance while walking at a comfortable pace. The walking distance was designed to be a total of 7 m, by adding 1.5 m from the start and the finish line as acceleration and deceleration zones [22,23,24]. 

The chair stand test involved measuring the time taken for the subject to rise 5 times, as fast as possible, from sitting in a chair to standing [25]. Isokinetic muscle strength in the quadriceps was measured using an isokinetic dynamometer (Biodex Medical Systems 3, Inc., Shirley, NY, USA) in the dominant leg. Before testing, the subjects performed a warm-up exercise consisting of 5 submaximal knee extensions of the leg at 60 degrees/second. In the main test session, subjects performed 5 isokinetic knee extensions of the leg at an angular velocity of 60 degrees/second, and subjects recovered passively for 2 min between sets of measurements [26,27]. The subject’s knee joint center, measured using the femoral epicondyles as a joint center reference, was aligned with the axis of the isokinetic dynamometer. The leg was secured against the knee attachment pad, while the trunk, waist, and mid-thigh were secured with straps to the chair to reduce movement of these segments.

### 2.3. Administration of the SC Supplements

SC in powdered form were provided by Bioport Korea Inc. (Gigang, Korea); the lot number (BPK-SC-001) was the same as that used in another animal study [28]. Simply, the collected SC (100 kg) was extracted in 20% ethanol at 90 °C for 4 h and filtered with an 80-mesh filter. The extract was lyophilised and yielded 23.2% reddish brown powders. The SC contained 4.92 ± 0.06 mg/g of schizandrin as a specific ingredient by high-performance liquid chromatography analysis [28].

The SC provided by the company has been proven effective through animal studies [3,4,13,28], and the safety of the human body has been confirmed by the Regional Clinical Trials Center, Dong-A University, Busan, Korea.

Previous animal studies showed that the optimal dose for the effects of SC extract on muscle strength was 250 mg/kg [13,28]. 

In this study, participant weight ranged from 49.4–68.0 kg. SC intake was based on the lowest body weight, and the value was about 1000 mg per day. The equation for human equivalent dose (HED) is as follows:

HED (mg/kg) = Animal does mg/kg km ratio [29].

All participants were randomized to receive placebo or SC pills containing 1000 mg of natural extract (twice per day, two pills per dose, each containing 250 mg). Participants were instructed to take each treatment pill as prescribed after 30 min of breakfast and dinner for 12 weeks and to complete a daily diary of compliance. All participants were trained to maintain their daily physical activity and advised to not consume other dietary supplements during the intervention. Patients and clinical personnel were blinded to the treatment assigned until all data was collected at the 12-week visit.

### 2.4. Biochemical Measurement

Fasting venous blood samples were collected from all participants, at baseline and at 12 weeks. Fasting was maintained for 12 h, and blood samples were collected between 8:00–9:00 a.m. on the following day. Plasma samples were obtained by centrifugation and stored at −80 °C. Serum creatine kinase (CK) and lactate dehydrogenase (LDH) were measured from serum frozen with a commercial kit, according to the manufacturer’s recommendations (Wiener Lab., Rosario, Argentina) [30]. Lactate and pyruvate concentration were determined by the fluorometric method of Maughan [31], which is based on the enzymatic reactions of NAD+ reduction (lactate) or NADH oxidation (pyruvate).

### 2.5. Statistical Analysis

The SPSS statistical package version 23.0 for Windows (SPSS, Inc., Chicago, IL, USA) was used to perform all statistical evaluations. We used two-factor repeated measures ANOVA to determine interaction (group × time) effects for all outcome variables. When significant group × time interactions were identified, paired t-tests were performed to detect differences between time points. Statistical significance was accepted at the *p* < 0.05 level. All variables are presented as the mean ± standard deviation.

## 3. Results

The mean SC extract supplementation and placebo compliance were 97.3% (90.2−100%) in the SC group and 95.8% (86.7–100%) in the placebo group. There were no significant differences between the groups for height, weight, or BMI at baseline (Table 1).

ASM/height^2^, grip strength, one-leg standing, 4 m walking, 5-times chair stand, CK, LDH, and pyruvate were measured at baseline and after 12 weeks. The results are presented in Table 2 and Table 3. No significant interactions were noted with time between the two groups with regards to ASM, physical function variables, and biomarkers.

QMS is presented in Figure 1. There was a significant interaction (*p* = 0.001) effect for group by time on QMS. While QMS significantly increased (*p* < 0.001) in the SC group from pre- to post-test, this value did not change in the placebo group.

Lactate level is presented in Figure 2. There was a significant interaction (*p* = 0.038) effect for group by time on lactate level. While lactate level significantly decreased (*p* = 0.045) in the SC group from pre- to post-test, this value did not change in the placebo group.

## 4. Discussion

Maintaining a high level of muscle strength can be an important means of preventing frailty and hospitalization, as well as physical disability and function limitation in IADLs [11,12,32,33]. Adequate nutritional supplementation is an important element of any strategy for the preservation of skeletal muscle mass, muscle strength, and muscle protein synthesis during aging [34,35,36]. SC is one of the most well-known herbal medicines and has been extensively used in Asia, including Japan, China, and Korea [37]. Nevertheless, human studies on the use of SC extract are limited. A randomized, double-blind, placebo-controlled study by Kim et al. reported that SC extract supplementation for 12 weeks was effective against menopausal symptoms, especially hot flushes, sweating, and heart palpitations [38]. However, another randomized, double-blind, placebo-controlled study of obese women by Song et al. reported that SC extract supplementation for 12 weeks had no effect on significant obesity-related parameters [2]. Therefore, the effect of SC extract on humans is not yet clear. Recently, two studies have shown that SC supplementation increased calf muscle strength as well as muscle mass in mice [13,28]. Our study, however, is the first to investigate the association between muscle strength and SC extract in humans. In addition, the results of our study showed that supplementation of SC extract for 12 weeks in humans improved muscle strength (mean 7.7% for QMS). Trapani et al. reported that the contractive strength of muscles is determined by the activity of adenosine triphosphatase (ATPase) [39]. In addition, SC extract supplementation has been shown to increase ATPase activity in gastrocnemius and soleus muscles in mice [13]. This may be the theoretical basis for the mechanism of increase in muscle strength by SC extract supplementation. However, our study did not investigate the activity of ATPase. This needs to be established through further study.

Skeletal muscle mass tended to increase in this study, but it was not statistically significant. A change in muscle strength is known to have a positive relationship with changes in muscle mass [40,41,42]. However, it is not a strong relationship, and individual differences may be present in adults [40,42]. In previous animal experiments with a positive effect on skeletal muscle and biomarkers, SC extract supplementation recommended a dose of 500 mg/kg (2430 mg/day when converted to 60 kg human) [13]. 

In our study, the SC extract supplement was 1000 mg per day. The SC extract we provided may not be sufficient enough to induce a positive effect on skeletal muscle metabolism [13].

In our study, ASM and other physical performance variables did not change after 12 weeks. A few studies reported that leg muscle strength is related to hand grip strength, gait speed, chair stand, and balance in adults [43,44,45]. In spite of this, our SC group had no changes in other physical performance variables, except leg muscle strength. This result differs from the increase in skeletal muscle found in mice after SC extract supplementation. Therefore, a clear relationship between physical performance variables and SC extract supplementation should be demonstrated through further studies.

Resting lactate level was measured at baseline and after 12 weeks, and results are presented in Figure 2. There was a significant interaction (*p* = 0.038) effect for group by time on resting lactate level. In the SC group, the lactate level at rest was significantly decreased (*p* < 0.05), but had no change in the placebo group after 12 weeks.

A follow-up study by Matsushita et al. reported that resting lactate levels were associated with body fat, blood pressure, serum lipids, insulin, and high sensitivity C-reactive protein in adults [20]. In other cross-sectional and follow-up studies, a high lactate level at rest was associated with chronic diseases such as type 2 diabetes and hypertension [19,46]. It was associated with a high risk of coronary heart disease (CHD), stroke, and all-cause mortality [20].

The normal range of lactic acid at rest is not yet clear. The normal range for lactate in the study of Matsushita et al. was 2.3–5.3 mg/dL [20], which was also similar in Juraschek et al.’s study [46]. However, in our subjects, the lactate level at rest was as high as 8.7 mg/dL.

Recently, animal study suggests that SC intake could lower the lactate level caused by exercise [15]. SC was also effective for menopausal symptoms in middle-aged women [38]. However, the effect of SC on lactic acid at rest was still unclear. Our study is the first to demonstrate the effect of SC supplementation on resting lactate. 

The exact mechanism to explain the effect of SC extract supplementation on the lactate level is not yet known. The results of Zagari et al. have shown the association between high lactate production and reduced oxidative metabolism [47]. They also reported that mitochondria play a central role in lactate metabolism [47]. An animal study by Chiu et al. reported that long-term SC treatment could enhance the renal mitochondrial antioxidant status, as well as improve mitochondrial function [48]. We thought of the possibility that the improvement of lactate metabolism may be due to the improvement of mitochondrial function by SC extract supplementation. Nevertheless, further studies are needed to clarify the association between SC supplementation and lactate concentration. CK catalyzes ATP dependent phosphorylation of creatine, and LDH is involved in the metabolism of pyruvate in skeletal muscle [49,50]. CK and LDH are reported to serve as markers of acute or chronic muscular damage and cell necrosis [51,52]. Additionally, activation of CK and LDH may increase symptoms such as pain, fatigue, and a decline in muscular strength during high-intensity, long-distance exercise due to damaged skeletal muscles [51,52]. Recently, in an animal study, SC was shown to effectively delay CK and LDH increases after acute exercise [13], but another animal study did not shown in this effect [53]. In our study, SC extract supplementation did not alter CK, LDH, or pyruvate at rest in humans.

Our research has some limitations. Firstly, participants’ physical activity and diet were not examined during the intervention period. We trained subjects to not to alter lifestyles (participation in physical activity programs, or intake of other healthy supplementation) that could affect the study during the intervention. Secondly, the subjects did not include men, so the effects of SC on muscle strength and resting lactic acid could not be seen in both men and women. Thirdly, the sample size was relatively small.

## 5. Conclusions

In conclusion, SC extract supplementation increased QMS and decreased lactate level at rest in adult women. SC extract is thought to be a health food that supports the healthy life of post-menopausal middle-aged women. Our results illustrate the potential of SC extract supplementation in the prevention of declining muscle strength. Further studies are required to cement this link and investigate whether this influences physical performance.

## Figures and Tables

**Figure 1 ijerph-17-02475-f001:**
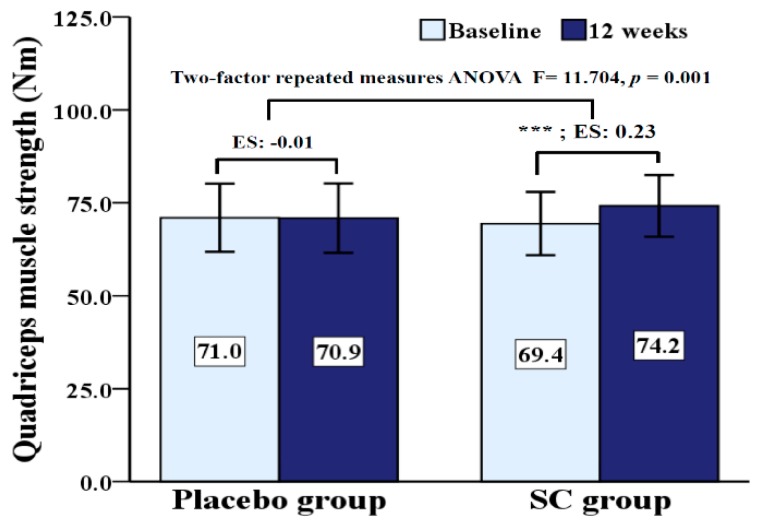
Comparisons of quadriceps muscles strength at baseline and 12 weeks after age adjustment. SC: *Schisandra chinensis*; ES: effect size; *** *p* < 0.001 versus baseline interventions.

**Figure 2 ijerph-17-02475-f002:**
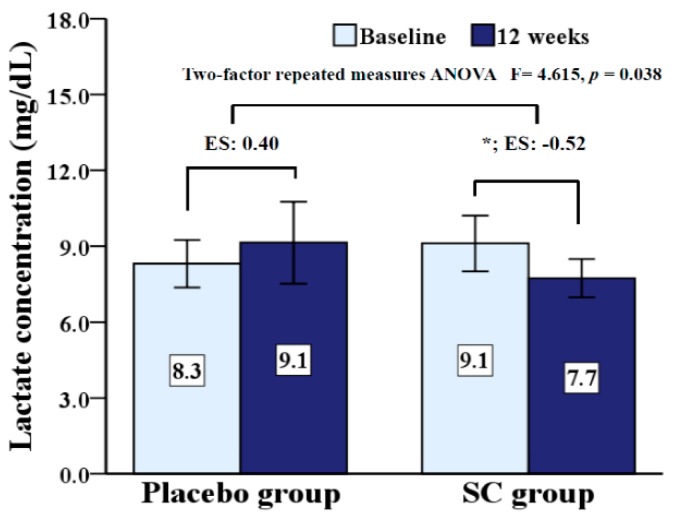
Comparisons of lactate concentration at baseline and 12 weeks after age adjustment. SC: *Schisandra chinensis*; ES: effect size; * *p* < 0.05 versus between baseline and after intervention.

**Table 1 ijerph-17-02475-t001:** The characteristics of the subjects.

Variables	SC Group	Placebo Group	*p*-Value
	(*n* = 24)	(*n* = 21)	
Age (years)	62.8 ± 6.5	63.4 ± 6.0	0.735
Height (cm)	156.2 ± 4.4	154.8 ± 3.9	0.178
Weight (kg)	57.3 ± 5.3	57.1 ± 4.9	0.831
Body mass index (kg/m^2^)	23.5 ± 2.3	23.9 ± 2.2	0.939

Values are means ± SD; SC: *Schisandra chinesis*.

**Table 2 ijerph-17-02475-t002:** Comparisons of appendicular skeletal mass and physical performance variables at baseline and at 12 weeks, after age adjustement.

Variable	Groups	Baseline	12 Weeks	*F*-Values	Group × Time Interaction *p*-Values
ASM/height^2^ (kg/m^2^)	Placebo group	6.2 ± 0.6	6.2 ± 0.6	3.616	0.064
	SC group	5.8 ± 0.5	5.9 ± 0.5		
Grip strength (kg)	Placebo group	21.7 ± 3.4	22.6 ± 4.5	0.007	0.935
	SC group	22.2 ± 4.0	22.0 ± 2.7		
One-leg standing (s)	Placebo group	17.6 ± 14.3	23.0 ± 26.6	0.688	0.411
	SC group	24.0 ± 18.3	35.3 ± 28.2		
4m walking time (s)	Placebo group	3.3 ± 0.4	3.0 ± 0.4	0.9260	0.341
	SC group	3.3 ± 0.4	3.1 ± 0.4		
5-times chair stand tests (s)	Placebo group	8.0 ± 0.9	7.4 ± 1.0	0.420	0.52
	SC group	7.6 ± 1.2	6.6 ± 1.3		

Values are means ± SD; SC: *Schisandra chinesis*; ASM: appendicular skeletal mass.

**Table 3 ijerph-17-02475-t003:** Comparisons of blood variables at baseline and 12 weeks, after age adjustment.

Variable	Groups	Baseline	12 Weeks	*F*-Values	Group × Time Interaction *p*-Values
Creatine kinase (U/L)	Placebo group	104.2 ± 26.6	100.1 ± 36.6	0.021	0.885
	SC group	89.0 ± 35.9	86.8 ± 36.5		
Lactate dehydrogenase (U/L)	Placebo group	390.1 ± 62.2	396.8 ± 50.8	0.106	0.747
	SC group	400.0 ± 88.1	415.3 ± 103.5		
Pyruvate (mg/dL)	Placebo group	0.62 ± 0.2	0.81 ± 0.2	0.133	0.717
	SC group	0.60 ± 0.2	0.74 ± 0.2		

Values are means ± SD; SC: *Schisandra chinesis*.

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
