# Peer review of "Effect of Schisandra chinensis Extract Supplementation on Quadriceps Muscle Strength and Fatigue in Adult Women: A Randomized, Double-Blind, Placebo-Controlled Trial"

_ijerph, 2020, doi:10.3390/ijerph17072475_

Round 1

Reviewer 1 Report

The manuscript is well written. The overall merit for the attempt is high, however, serious concerns should be addressed:

The experiment should be replicated in order to guarantee the findings. Sample is low, do not contain men, and is very heterogeneous, as authors have stated. Authors should describe the age range of women due to it is important in order to guarantee a similar hormonal status in the study sample (i.e., are all participants post-menopausal women?)

The day time in which training and test of each group were developed should be described. Day time affects performance and motor learning, as well as the female hormonal status according to menstrual cycle stage (if appropriate).

Details of the source of Schisandra chinensis should be provided (commercial brand, origin…). The criteria for the amount of Schisandra chinensis administration should also be improved, in order to establish the minimum dose that produces performance enhancement. Patient compliance with the treatment should be also addressed.

Inflammatory and muscle damage markers are scarce. There are markers that could provide better evidence for the aim of this study, in the long and the short term (c-reactive protein, ammonia, endogenous antioxidants, DHA, cortisol…)

Performance test could be also improved. Recent studies employ a lineal encoder in order to evaluate power loss as a result of muscle fatigue, using the force-velocity ratio as state of the art and one of the best markers of muscle functioning.

Minor spelling amendments are desirable.

Author Response

For Reviewer

The manuscript is well written. The overall merit for the attempt is high, however, serious concerns should be addressed:

Thank you for your valuable and appropriate comments. We have revised for your valuable and appropriate comments as follows:

  1. The experiment should be replicated in order to guarantee the findings. Sample is low, do not contain men, and is very heterogeneous, as authors have stated. Authors should describe the age range of women due to it is important in order to guarantee a similar hormonal status in the study sample (i.e., are all participants post-menopausal women?)

Thank you for your valuable and appropriate comments.

Despite your appropriated comments, we reviewed previous studies to derive an adequate number of samples and found that the 8-week plant extract (Withania somnifera) supplementation effect size on muscle strength (leg extension) of the lower body in a study of adult males was 0.5 [1].

And, in another study, the size of the effects of plant extract (Panax Notoginseng) supplementation on physical function for 30 days ranged from 0.32 – 0.62 [2]. In this study, 34 samples were required when the effect size was 0.32. In our study, 45 people completed the experiment and met the required number of samples. Nevertheless, we appreciate the appropriate comments on the reviewer's sample size. When interpreting the results, we should consider the limitations of our research.

Results of G* power test

F tests - ANOVA: Repeated measures, within-between interaction

Analysis:   A priori: Compute required sample size

Input:         Effect size f      =            0.32

      α err prob         =           0.05

      Power (1-β err prob)    =            0.95

      Number of groups       =            2

      Number of measurements       =           2

      Corr among rep measures       =           0.5

      Nonsphericity correction ε      =           1

Output:      Noncentrality parameter λ      =           13.9264000

      Critical F           =           4.1490974

      Numerator df  =           1.0000000

      Denominator df            =            32.0000000

      Total sample size         =            34

      Actual power   =           0.9513089

In addition, all participants are postmenopausal middle-aged women. Therefore, we have modified the text a commenter.

  1. The day time in which training and test of each group were developed should be described. Day time affects performance and motor learning, as well as the female hormonal status according to menstrual cycle stage (if appropriate).

Thank you for your valuable and appropriate comments.

The physical function test used the maximum value after taking measurements of four different physical parameters twice between am 8 - 10 times.

 Participants took each treatment pill as prescribed after 30 minutes of breakfast and dinner for 12 weeks.

Fasting was maintained for 12 h, and blood samples were collected between 8-9 am on the following day.

Therefore, we added to the text as a commenter.

  1. Details of the source of Schisandra chinensis should be provided (commercial brand, origin…). The criteria for the amount of Schisandra chinensis administration should also be improved, in order to establish the minimum dose that produces performance enhancement. Patient compliance with the treatment should be also addressed.

Thank you for your valuable and appropriate comments. The manufacturers and production methods of Schisandra Chinensis (SC) are as follows and added to the text as a commenter.

SC is a health supplement available in various formulations. SC in the powdered form was provided by Bioport Korea Inc. (Gigang, Korea); the lot number (BPK-SC-001) was the same as that used in another animal study [3].

And, we have improved the information about SC intake in the text, as reviewers comment. Also, we showed the compliance of the participants with the SC intake in the first paragraph of results section.

Simply, the collected Schisandrae Fructus (100 kg) was extracted in 20% ethanol at 90 °C for 4 h, and filtered with an 80-mesh filter. The extract was lyophilized and yielded 23.2% reddish-brown powders. The SC contained 4.92±0.06 mg/g of schizandra as a specific ingredient by high-performance liquid chromatography analysis [3] .

The SC provided by the company has been proven effective through animal studies [3, 4], and the safety of the human body has been confirmed by the Regional Clinical Trials Center, Dong-A University Medical center, Busan, Korea.

  1. Inflammatory and muscle damage markers are scarce. There are markers that could provide better evidence for the aim of this study, in the long and the short term (c-reactive protein, ammonia, endogenous antioxidants, DHA, cortisol…)

Your comment is very appropriate, and as you know, there are various studies on inflammatory and muscle damage markers in the field of aging science. In spite of your point of view, the markers we assessed are also very rare studies with SC and important markers, in particular, creatine kinase (CK) and lactate dehydrogenase (LDH) are reported as representative muscle damage markers [3-7]. And, animal studies have confirmed that SC intake delays the increase of exercise-induced muscle damage markers such as CK and LDH [3, 4]. Therefore, we examined whether SC supplements could regulate muscle damage markers in humans. In this study, inflammatory markers and endogenous antioxidants were not examined, which is considered to be an additional limitation of the study.

  1. Performance test could be also improved. Recent studies employ a lineal encoder in order to evaluate power loss as a result of muscle fatigue, using the force-velocity ratio as state of the art and one of the best markers of muscle functioning.

Thank you for your valuable and appropriate comments.

Like the reviewer's comment, in recent research, a linear encoder is used to test the muscle force-velocity ratio [8-10]. The femoral strength evaluation using the isokinetic strength mechanism has the advantages of high reliability and small measurement error [11]. It is also known that it can be used for clinical evaluation because of the high accuracy of the results observed through the test [11]. Therefore, isokinetic muscle strength in the quadriceps was measured using an isokinetic dynamometer (Biodex Medical Systems 3, Inc., Shirley, NY, USA).

  1. Minor spelling amendments are desirable.

Thank you for your valuable and appropriate comments. We have corrected some minor spelling as your comment.

Reviewer 2 Report

Specific concerns: Section 2.3. The stated dosage range from animal studies was 125-250 mg/kg. Using a mean body weight of 60 kg, the dose given was 1000 mg/day, or 16.7 mg/kg. The discrepancy needs to be explained.

Section 2.4 The blood sample preparation described appears to be for plasma, not serum. Does this affect the test results?

Figure 1 legend Se used instead of ES as in Figure itself and in Figure 2.

In the discussion section it would be helpful to give the range of lactate concentration that is considered as clinically normal. The observed results were not particularly large in absolute magnitude, additional context for their significance should be provided.

Broad concern: The active ingredient in SC extract is not given (or not known). The manufacturer of the extract used is also not given. Sadly, nutraceutical suppliers have developed a reputation for poor quality control, frequently delivering in the finished product inaccurate extract weights that are often degraded and no longer contain much of the active compound. For a reviewed scientific paper it is critical to provide sufficient information on the content of the supplement and the reproducibility from lot to lot of the samples used. Some stability studies of the encapsulated extract would also be helpful. Without any quality data the clinical results seen mean little and inspire no confidence they can be reproduced.

Author Response

For Reviewer 2

Comments and Suggestions for Authors

Thank you for your valuable and appropriate comments. We have revised for your valuable and appropriate comments as follows:

  1. Specific concerns: Section 2.3. The stated dosage range from animal studies was 125-250 mg/kg. Using a mean body weight of 60 kg, the dose given was 1000 mg/day, or 16.7 mg/kg. The discrepancy needs to be explained.

Your suggestion is appropriate. There was a mistake in the manuscript.

Therefore, we have modified the text as follows:

Previous animal studies showed that the minimum dose for the effects of SC extract on muscle strength was 250 mg/kg. In this study, participant weight range was 49.4 – 68.0 kg. SC intake was based on the lowest body weight and the value was about 1,000 mg per day. The equation for the human equivalent dose (HED) is as follows:

HED (mg/kg) = Animal does mg/kgâ…¹Km ratio [12]

HED (mg/kg) = 250mg/kgⅹ(3/37)≒ 20.3 mg/kg

In case of 49kg human, real dose= HED (mg/kg)â…¹body weight (kg)

20.3 mg/kgⅹ49.4≒ 1,000 mg (1002.8 mg)/49.4kg

  1. Section 2.4 The blood sample preparation described appears to be for plasma, not serum. Does this affect the test results?

Your suggestion is appropriate. We have revised the text.

  1. Figure 1 legend Se used instead of ES as in Figure itself and in Figure 2.

Your suggestion is appropriate. We have revised the text.

  1. In the discussion section it would be helpful to give the range of lactate concentration that is considered as clinically normal. The observed results were not particularly large in absolute magnitude, additional context for their significance should be provided.

Thank you for your valuable and appropriate comments. We made the following modifications:

A follow-up study by Matsushita et al. reported that resting lactate concentrations were associated with body fat, blood pressure, serum lipids, insulin, and high sensitivity C-reactive protein in adults [13]. In other cross-sectional and follow-up studies, high lactate concentration at rest was associated with chronic diseases such as type 2 diabetes and hypertension [14, 15]. It was associated with a high risk of coronary heart disease (CHD), stroke, and all-cause mortality [13].

The normal range of lactic acid at rest is not yet clear. The normal range for lactate in the study of Matsushita et al. was 2.3-5.3 mg / dl [13], which was also similar in Juraschek et al.'s study [15]. However, in our subjects, the lactate concentration at rest was high as 8.7 mg/dl.

Recently, Animal study suggests that SC intake could lower the lactate level caused by exercise [16]. SC was also effective for menopausal symptoms in middle-aged women [17]. However, the effect of SC on lactic acid at rest was still unclear. Our study is the first to demonstrate the effect of SC supplementation on resting lactate.

The exact mechanism to explain the effect of SC extract supplementation on lactate concentration is not yet known. The results of Zagari et al. have shown the association between high lactate production and reduced oxidative metabolism [44]. They also reported that mitochondria play a central role in lactate metabolism [44]. Animal study by Chiu et al. reported that long-term SC treatment could enhance renal mitochondrial antioxidant status as well as improve mitochondrial functional [45]. Therefore, we believe that the improvement of lactate metabolism due to the improvement of mitochondrial function by SC extract supplementation the lactate at rest.

  1. Broad concern: The active ingredient in SC extract is not given (or not known). The manufacturer of the extract used is also not given. Sadly, nutraceutical suppliers have developed a reputation for poor quality control, frequently delivering in the finished product inaccurate extract weights that are often degraded and no longer contain much of the active compound. For a reviewed scientific paper it is critical to provide sufficient information on the content of the supplement and the reproducibility from lot to lot of the samples used. Some stability studies of the encapsulated extract would also be helpful. Without any quality data the clinical results seen mean little and inspire no confidence they can be reproduced.

Your suggestion is appropriate.

SC in powdered form were provided by Bioport Korea Inc. (Gigang, Korea); the lot number (BPK-SC-001) was the same as that used in other animal study [3].

The SC provided by the company has been proven effective through animal studies [3, 4], and the safety of the human body has been confirmed by the Regional Clinical Trials Center, Dong-A University, Busan, Korea.

Reviewer 3 Report

I think it would help to have the acronyms (e.g. hematocrit, hemoglobin etc) written out in the first instance you use them in the body of the paper.

In the section Physical Function test- 

You mention that 4 tests were used, yet 5 are listed (grip strength, one leg standing test, walking time, chair stand and isokinetic muscle strength). 

The walking time test is poorly defined. What are the acceleration/ deceleration zones? 

In the Results Section-

Figure 1. Caption, change to ES from SE

Table 2. Caption should read "... variables at baseline and at 12 weeks, following adjustment for age"

Line 137-138: Reword this " there were no significant differences between the groups on height, weight or BMI at baseline" 

I would prefer to see the beta values also for the ANOVAs. 

You haven't explained your lactate findings shown in Fig2 in the Results section, and these values don't match the lactate values displayed in Table 3, in fact it shows that lactate actually increased in the table (400 to 415.3 at 12 weeks in the SC group).  

In the Discussion: 

Line 197-198: You wrote that the lactate concentration was significantly decreased (p<0.05), but not statistically significant. Consider re-wording as this is contradictory. 

You also write that the placebo group showed a trend. Please describe what the trend was toward e.g. a trend toward increased lactate was observed in the placebo group.

Line 224: pyruvate, not pyruvic

Author Response

For Reviewer 3

Comments and Suggestions for Authors

Thank you for your valuable and appropriate comments. We have revised for your valuable and appropriate comments as follows:

  1. I think it would help to have the acronyms (e.g. hematocrit, hemoglobin etc) written out in the first instance you use them in the body of the paper.

Your suggestion is appropriate. We have revised the text

  1. In the section Physical Function test-

You mention that 4 tests were used, yet 5 are listed (grip strength, one leg standing test, walking time, chair stand and isokinetic muscle strength).

Thank you for your valuable and appropriate comments.

We revised the text as a commenter.

  1. The walking time test is poorly defined. What are the acceleration/ deceleration zones?

Thank you for your valuable and appropriate comments. We modified as a commenter as follows:

The walking test measured the time taken to walk a 4m straight distance with walk at comfortable pace. The walking distance was designed to walk a total of 7 m by adding 1.5 m from the starting and the finish line for acceleration and deceleration zone.

In the Results Section-

  1. Figure 1. Caption, change to ES from SE

Your suggestion is appropriate. We have revised the text.

  1. Table 2. Caption should read "... variables at baseline and at 12 weeks, following adjustment for age"

Your suggestion is appropriate. We have revised the text.

  1. Line 137-138: Reword this " there were no significant differences between the groups on height, weight or BMI at baseline"

Your suggestion is appropriate. We have revised the text as a commenter (Line 158-159).

  1. I would prefer to see the beta values also for the ANOVAs.

We present the F value for analysis of variance in the table of analysis results.

  1. You haven't explained your lactate findings shown in Fig2 in the Results section, and these values don't match the lactate values displayed in Table 3, in fact it shows that lactate actually increased in the table (400 to 415.3 at 12 weeks in the SC group). 

Your suggestion is appropriate. We have revised the text as a commenter. We have missed filling out for lactate concentration in the results section. Therefore, added result for lactate concentration in results section.

In the Discussion:

  1. Line 197-198: You wrote that the lactate concentration was significantly decreased (p<0.05), but not statistically significant. Consider re-wording as this is contradictory.

Your suggestion is appropriate. We have revised the text as a commenter.

  1. You also write that the placebo group showed a trend. Please describe what the trend was toward e.g. a trend toward increased lactate was observed in the placebo group.

.

Thank you for your valuable and appropriate comments. We have revised the text as a commenter.

  1. Line 224: pyruvate, not pyruvic

Your suggestion is appropriate. We have revised the text as your comment.

Reviewer 4 Report

This manuscript evaluated the effect of Schisandra chinensis extract supplementation on quadriceps muscle strength and lactate at rest in adult women. Forty-five healthy adult women (61.9 ± 8.4 years) were randomly divided into the SC (n = 24) or the placebo group (n = 21). The SC group consumed 1,000 mg of SC extract per day, whereas the placebo group consumed 1,000 mg of starch per day for 12 weeks. The difference in muscle mass, physical function, and bio-markers and the relative changes between baseline and 12 weeks were evaluated. The results found that supplementation of SC extract may help to improve quadriceps muscle strength as well as decrease lactate concentration at rest in adult women.

        Even though the results of the clinical tests were interesting, however, the manuscript needs substantial revision before it will be recommended for publication in the journal.

  1. The main shortage of the manuscript is that the authors did not mention how the Schisandra chinensis extract was prepared or provided.
  2. The chemical composition of water extract, ethyl alcohol extract or chloroform extract vary dramatically. The solubility of the lignin ingredients in Schisandra chinensis such as of schisandrin, schisandrol B, deoxyschisandrin, gomisin N, gamma-schizandrin and schisandrin C in solvents with different polarity differ dramatically. In view of this, the authos are recommended to provive the HPLC chromatogram showing the fingerprint of the Schisandra chinensis extract studied.
  3. The Latin names of the plant species, Schisandra chinensis, should be italicized.
  4. Please go through the manuscript including the references section and revise accordingly.
  5. In the abstract, “wherease” should be corrected as “whereas”

Author Response

For Reviewer 4

Comments and Suggestions for Authors

This manuscript evaluated the effect of Schisandra chinensis extract supplementation on quadriceps muscle strength and lactate at rest in adult women. Forty-five healthy adult women (61.9 ± 8.4 years) were randomly divided into the SC (n = 24) or the placebo group (n = 21). The SC group consumed 1,000 mg of SC extract per day, whereas the placebo group consumed 1,000 mg of starch per day for 12 weeks. The difference in muscle mass, physical function, and bio-markers and the relative changes between baseline and 12 weeks were evaluated. The results found that supplementation of SC extract may help to improve quadriceps muscle strength as well as decrease lactate concentration at rest in adult women.

Even though the results of the clinical tests were interesting, however, the manuscript needs substantial revision before it will be recommended for publication in the journal.

Thank you for your valuable and appropriate comments. We have revised for your valuable and appropriate comments as follows:

  1. The main shortage of the manuscript is that the authors did not mention how the Schisandra chinensis extract was prepared or provided.

The chemical composition of water extract, ethyl alcohol extract or chloroform extract vary dramatically. The solubility of the lignin ingredients in Schisandra chinensis such as of schisandrin, schisandrol B, deoxyschisandrin, gomisin N, gamma-schizandrin and schisandrin C in solvents with different polarity differ dramatically. In view of this, the authos are recommended to provive the HPLC chromatogram showing the fingerprint of the Schisandra chinensis extract studied.

Thank you for your valuable and appropriate comments.

SC in powdered form were provided by Bioport Korea Inc. (Gigang, Korea); the lot number (BPK-SC-001) was the same as that used in other animal study [3].

Simply, the collected SC ex (100 kg) was extracted in 20% ethanol at 90 °C for 4 h, and filtered with a 80-mesh filter. The extract was lyophilised and yielded 23.2% reddish brown powders. The SC contained 4.92±0.06 mg/g of schizandrin as a specific ingredient by high-performance liquid chromatography analysis [3] .

Figure 1. HPLC chromatogram of extract of Schisandra chinensis. The analysis conditions were: column, Symmetry C18 4.6 mm 250 mm (Waters Corporation); M. phase, 70 % ACN (0.1 %F. A); flow rate, 0.6 ml/min; inject volume, 10 ul; detector, PDA 254 nm.

  1. The Latin names of the plant species, Schisandra chinensis, should be italicized.

Your suggestion is appropriate. We have revised the text as a commenter

  1. Please go through the manuscript including the references section and revise accordingly.

We have revised the abstract as follows.

  1. In the abstract, “wherease” should be corrected as “whereas”

Your suggestion is appropriate. We have revised the abstract as your suggestion.

Reference

  1. Wankhede S, Langade D, Joshi K, Sinha SR, Bhattacharyya S. Examining the effect of Withania somnifera supplementation on muscle strength and recovery: a randomized controlled trial. J Int Soc Sports Nutr 2015;12:43.
  2. Liang MT, Podolka TD, Chuang WJ. Panax notoginseng supplementation enhances physical performance during endurance exercise. Journal of Strength and Conditioning Research 2005;19(1):108-114.
  3. Kim JW, Ku SK, Kim KY, Kim SG, Han MH, et al. Schisandrae Fructus Supplementation Ameliorates Sciatic Neurectomy-Induced Muscle Atrophy in Mice. Oxid Med Cell Longev 2015;2015:872428.
  4. Kim KY, Ku SK, Lee KW, Song CH, An WG. Muscle-protective effects of Schisandrae Fructus extracts in old mice after chronic forced exercise. J Ethnopharmacol 2018;212:175-187.
  5. Zhang Y, Huang JJ, Wang ZQ, Wang N, Wu ZY. Value of muscle enzyme measurement in evaluating different neuromuscular diseases. Clin Chim Acta 2012;413(3-4):520-524.
  6. Choi M, Park H, Cho S, Lee M. Vitamin D3 supplementation modulates inflammatory responses from the muscle damage induced by high-intensity exercise in SD rats. Cytokine 2013;63(1):27-35.
  7. Farias-Junior LF, Browne RAV, Freire YA, Oliveira-Dantas FF, Lemos T, et al. Psychological responses, muscle damage, inflammation, and delayed onset muscle soreness to high-intensity interval and moderate-intensity continuous exercise in overweight men. Physiol Behav 2019;199:200-209.
  8. van den Tillaar R, Ball N. Validity and Reliability of Kinematics Measured with PUSH Band vs. Linear Encoder in Bench Press and Push-Ups. Sports (Basel) 2019;7(9).
  9. Lindemann U, Farahmand P, Klenk J, Blatzonis K, Becker C. Validity of linear encoder measurement of sit-to-stand performance power in older people. Physiotherapy 2015;101(3):298-302.
  10. Jimenez MF, Laverty TM, Bombaci SP, Wilkins K, Bennett DE, et al. Underrepresented faculty play a disproportionate role in advancing diversity and inclusion. Nat Ecol Evol 2019;3(7):1030-1033.
  11. Eitzen I, Hakestad KA, Risberg MA. Inter- and intrarater reliability of isokinetic thigh muscle strength tests in postmenopausal women with osteopenia. Arch Phys Med Rehabil 2012;93(3):420-427.
  12. Nair AB, Jacob S. A simple practice guide for dose conversion between animals and human. J Basic Clin Pharm 2016;7(2):27-31.
  13. Matsushita K, Williams EK, Mongraw-Chaffin ML, Coresh J, Schmidt MI, et al. The association of plasma lactate with incident cardiovascular outcomes: the ARIC Study. American Journal of Epidemiology 2013;178(3):401-409.
  14. Crawford SO, Hoogeveen RC, Brancati FL, Astor BC, Ballantyne CM, et al. Association of blood lactate with type 2 diabetes: the Atherosclerosis Risk in Communities Carotid MRI Study. Int J Epidemiol 2010;39(6):1647-1655.
  15. Juraschek SP, Bower JK, Selvin E, Subash Shantha GP, Hoogeveen RC, et al. Plasma lactate and incident hypertension in the atherosclerosis risk in communities study. Am J Hypertens 2015;28(2):216-224.
  16. Cao S, Shang H, Wu W, Putheti R. Evaluation of anti-athletic fatigue activity of Schizandra chinensis aqueous extracts in mice. African Journal of Pharmacy and Pharmacology 2009;3(11):593-597.
  17. Park JY, Kim KH. A randomized, double-blind, placebo-controlled trial of Schisandra chinensis for menopausal symptoms. Climacteric 2016;19(6):574-580.
  18. Kim HJ, Park I, Lee HJ, Lee O. The reliability and validity of gait speed with different walking pace and distances against general health, physical function, and chronic disease in aged adults. J Exerc Nutrition Biochem 2016;20(3):46-50.
  19. Van Ancum JM, van Schooten KS, Jonkman NH, Huijben B, van Lummel RC, et al. Gait speed assessed by a 4-m walk test is not representative of daily-life gait speed in community-dwelling adults. Maturitas 2019;121:28-34.
  20. Ng SS, Au KK, Chan EL, Chan DO, Keung GM, et al. Effect of acceleration and deceleration distance on the walking speed of people with chronic stroke. J Rehabil Med 2016;48(8):666-670.

Round 2

Reviewer 2 Report

Thank you for addressing many of the concerns raised. However, the major concern of what is in the supplement and lot to lot variability has not been adequately addressed. Reference 29 addresses the extraction method but contains no chromatography or compound identification. An apt reference is Anal. Methods, 2014,6, 5981-5985 by where 2 schisandrols and 3 schisandrins were found in varying amounts; this may not include all components. Strikingly, significant differences were found in the relative amounts of each over the 4 months examined. Until demonstrated otherwise, it must be assumed that each of the five analytes has different bioactivity. In this instance, that the same lot of material was used in previous studies is a negative factor. To be minimally acceptable the amount of at least these five in the capsule must be given; without knowing which is the active ingredient or the lot to lot variation with this extraction protocol it is still not of much clinical significance.

The lactate results remain problematic as well. The lactate levels in the control at 12 weeks and the treated group baseline are identical, while the control baseline and treated 12 weeks are not significantly different. This suggests either a test with low precision/accuracy, or that natural variations in lactate are dominant. Neither possibility justifies the claim that schisandriin extract is relevant for controlling lactate levels.

Author Response

Thank you for addressing many of the concerns raised. However, the major concern of what is in the supplement and lot to lot variability has not been adequately addressed. Reference 29 addresses the extraction method but contains no chromatography or compound identification. An apt reference is Anal. Methods, 2014,6, 5981-5985 by Yong-Gang Xia,‡a   Bing-You Yang,‡a   Jun Liang,a   Jie-Shu Wanga  and  Hai-Xue Kuang*a  where 2 schisandrols and 3 schisandrins were found in varying amounts; this may not include all components. Strikingly, significant differences were found in the relative amounts of each over the 4 months examined. Until demonstrated otherwise, it must be assumed that each of the five analytes has different bioactivity. In this instance, that the same lot of material was used in previous studies is a negative factor. To be minimally acceptable the amount of at least these five in the capsule must be given; without knowing which is the active ingredient or the lot to lot variation with this extraction protocol it is still not of much clinical significance.

Thank you for your valuable and appropriate comments.

Schizandrin was established as an ingredient for quality control of raw materials, this fulfilled the requirements outlined in Korea's health functional food laws and regulation. It is also possible to set an index component with other component or a plurality of component, it was acknowledged that one type of schizandrin was sufficient.

When setting the indicator component standard, it was based on the results of three repeated tests (a total of 9 times) for 3 lot products from an accredited testing agency designated by the Ministry of Food and Drug Safety, Republic of Korea.

The specified indicator component has a standard of 5,0mg ±20%, which is recognized as the same raw material if it is met when manufacturing the raw material.

The Schisandra used in this study was cultivated in Mungyeong City and is the same raw material used in previous studies [1-3]. Therefore, we regard it as the same one. In addition, we present HPLC chromatogram and product test report.  

Reviewer 4 Report

Most queries raised in previous review had been satisfactorily addressed and revised.  The manuscript is recommended for publication in the Journal.

YS

Author Response

The lactate results remain problematic as well.

The lactate levels in the control at 12 weeks and the treated group baseline are identical, while the control baseline and treated 12 weeks are not significantly different. This suggests either a test with low precision/accuracy, or that natural variations in lactate are dominant. Neither possibility justifies the claim that schisandrin extract is relevant for controlling lactate levels.

Thank you for your valuable and appropriate comments.

Based on the effects of SC on lactic acid identified in previous animal experiments and the results confirmed in our study[4],

We thought SC supplementation might be involved in lactic acid control.

However, there is very little research to explain the relationship between SC supplementation and lactate concentration. Therefore, it is considered that additional research is needed to support our study.

We made some changes to the discussion.